# The Effect of Nerolidol on Renal Dysfunction following Bilateral Ureteral Obstruction

**DOI:** 10.3390/biomedicines12102285

**Published:** 2024-10-09

**Authors:** Harun R. Toumi, Sundus M. Sallabi, Loay Lubbad, Suhail Al-Salam, Fayez T. Hammad

**Affiliations:** 1Department of Surgery, College of Medicine & Health Sciences, United Arab Emirates University, Al Ain P.O. Box 17666, United Arab Emirates; 201640038@uaeu.ac.ae (H.R.T.); 201640052@uaeu.ac.ae (S.M.S.); loay_lubbad@uaeu.ac.ae (L.L.); 2Department of Pathology, College of Medicine & Health Sciences, United Arab Emirates University, Al Ain P.O. Box 17666, United Arab Emirates; suhaila@uaeu.ac.ae

**Keywords:** bilateral ureteral obstruction, nerolidol, acute kidney injury

## Abstract

**Background/Objectives:** Obstructive uropathy is a common cause of renal impairment. Recently, there has been a burgeoning interest in exploring natural products as potential alternative remedies for many conditions due to their low toxicity, affordability and wide availability. **Methods:** We investigated the effect of nerolidol in a rat model of bilateral ureteral obstruction (BUO) injury. Nerolidol, dissolved in a vehicle, was administered orally as a single daily dose of 200 mg/kg to Wistar rats. Sham group (n = 12) underwent sham surgery, whereas the BUO (n = 12) and BUO/NR groups (n = 12) underwent reversible 24-h BUO and received the vehicle or nerolidol, respectively. The treatment started 9 days prior to the BUO/sham surgery and continued for 3 days after reversal. Renal functions were assessed before starting the treatment, just prior to the intervention and 3 days after BUO reversal. **Results:** Neither nerolidol nor the vehicle affected the basal renal functions. Nerolidol resulted in a significant attenuation in the BUO-induced alterations in renal functional parameters such as serum creatinine and urea, creatinine clearance and urinary albumin-creatinine ratio. Nerolidol also attenuated the changes in several markers associated with renal injury, inflammation, apoptosis and oxidative stress and mitigated the histological alterations. **Conclusions:** The findings of this study demonstrated the potent reno-protective effects of nerolidol in mitigating the adverse renal effects of bilateral ureteral obstruction. This is attributed to its anti-inflammatory, anti-fibrotic, anti-apoptotic and anti-oxidant properties. These effects were reflected in the partial recovery of renal functions and histological features. These findings may have potential therapeutic implications.

## 1. Introduction

Obstructive uropathy stands as a significant contributor to acute kidney injury. It accounts for approximately 5–10% of all acute kidney injury causes [1]. If left untreated, acute kidney injury caused by obstructive uropathy can progress to renal failure, posing a grave risk of mortality [2]. The severity of kidney injury ensuing from obstructive uropathy depends on several factors, such as the unilateral or bilateral nature of the obstruction, whether the obstruction is complete or partial, and the duration of obstruction [3,4,5,6]. The treatment typically involves the removal of the obstruction using various interventional techniques [7]. Various medications have also been employed to counteract the alterations in kidney functions associated with urinary obstruction [7].

Recently, there has been growing interest in utilizing natural products as alternative remedies for several conditions and diseases, including obstructive uropathy, due to their low toxicity, affordability, and wide availability [8]. Nerolidol (3,7,11-trimethyl-1,6,10-dodecatrien-3-ol), also known as peruviol, is a naturally occurring sesquiterpene alcohol characterized by a strong floral odor. It is commonly found in the essential oils of several plants, such as *Piper claussenianum* [9,10]. It plays an important role in the defense mechanisms of many plants [11]. Nerolidol has been gaining a lot of attention lately because it has been proven to be effective in several conditions in various organs. For instance, nerolidol has been shown to exhibit neuroprotective effects in mouse hippocampus, as it mitigated the neurological damage induced by oxidative stress [12]. It has also been proven to have a cardioprotective effect in rats with myocardial infarction [13]. In addition, nerolidol has an antimicrobial effect against several microbial organisms including *Staphylococcus aureus*, *Escherichia coli*, and *Candida albicans* [14].

Owing to its pharmacological and biological activities, nerolidol has shown promise in treating various conditions affecting the kidney. For instance, nerolidol demonstrated therapeutic potential against lipopolysaccharide-induced acute kidney injury in rats [15]. The administration of nerolidol led to a reduction in kidney tissue injury markers and mitigation of the alterations in the pro-inflammatory cytokines, and ultimately resulted in an improvement in renal functions. In another in vivo and in vitro study, nerolidol exhibited potent protective effects against cyclophosphamide-induced renal toxicity [16]. It effectively mitigated changes in oxidative stress and apoptotic, inflammatory, and fibrotic markers. In addition, in a rat model, we have recently investigated the effect of nerolidol on renal ischemia–reperfusion injury [17]. The findings suggested that a daily dose of 200 mg/kg of nerolidol had significantly attenuated the ischemia–reperfusion injury-induced changes in the level of serum creatinine and urea, creatinine clearance, and urine albumin–creatinine ratio. Additionally, nerolidol reduced the expression of several kidney injury, pro-inflammatory, profibrotic, proapoptotic, and oxidative stress markers, as well as histological alterations. In view of this evidence, it is possible that nerolidol has a protective effect on ureteral obstruction. However, this effect has not been investigated yet. Therefore, the aim of this study was to investigate the effect of nerolidol in a rat model of reversible bilateral ureteral obstruction (BUO).

## 2. Materials and Methods

Studies were performed on male Wistar rats weighing between 220 and 250 g at the time of BUO. The rats were fed a normal diet consisting of standard rat chow. They were housed in standard cages, maintained on a 12 h light–dark cycle at a temperature of 22–24 °C. All of the rats were fasted overnight before surgery and other experimental procedures, but they had unlimited access to water. The experimental protocol was approved by the local ethics committee (ERA_2023_2387).

### 2.1. Bilateral Ureteral Obstruction and Reversal of Obstruction

All surgical procedures were conducted under strict aseptic conditions as previously detailed [18,19,20,21]. In brief, all of the rats were anesthetized via an intraperitoneal injection of ketamine hydrochloride (30 mg/kg) (Troy Animal Healthcare, Glendenning, NSW, Australia), followed by pentobarbital (45 mg/kg) (Jurox, Rhodes, NSW, Australia). A midline abdominal incision was made to expose the ureters of both kidneys. A 3–4 mm long bisected PVC tube (0.58 mm internal diameter) was wrapped around the mid-portion of both ureters. The PVC tubes were then tightened using a 4-0 silk suture to cause complete occlusion of the ureters without damaging the ureteral wall. Following the surgical procedure, the abdominal incision was closed with 4-0 vicryl suture, and the skin was closed using surgical staples. After 24 h, the obstructing silk sutures and PVC tubes were carefully removed, and the incision was closed as previously described. The unrestricted movement of urine through the previously obstructed area indicated a complete release of the obstruction.

### 2.2. Nerolidol and Vehicle Administration

Nerolidol (Sigma-Aldrich, St. Louis, MO, USA), a racemic mixture, was first dissolved in 0.5 mL of corn oil, which was used as a vehicle. The vehicle and the treatment were administered via an oral gavage needle as a single daily dose of 200 mg/kg for a period of 14 consecutive days. The dosage was selected based on previous studies [17]. G-sham and G-BUO received the vehicle only, whereas the G-BUO/NR group received the nerolidol–vehicle solution. The treatment for all groups started 9 days prior to the BUO surgery and continued all the way up to organ collection 3 days following BUO reversal. None of the treated animals showed any adverse effects.

### 2.3. Experimental Groups

The rats were assigned randomly to three groups:G-sham (n = 12): rats that underwent sham manipulation of both ureters and received the vehicle only.G-BUO (n = 12): rats that underwent bilateral ureteral obstruction for 24 h and received only the vehicle.G-BUO/NR (n = 12): rats that underwent bilateral ureteral obstruction for 24 h and received nerolidol dissolved in the vehicle.

### 2.4. Experimental Protocol and Sample Collection

Urine was collected using metabolic cages for 24 h at three different time points (Figure 1): just before the start of nerolidol/vehicle treatment for baseline pre-treatment values (basal), day 9 after nerolidol/vehicle treatment for pre-BUO values (pre-BUO), and on the 3rd day after the reversal of BUO (post-BUO), and the volumes of daily urine at these time points were calculated. Using the tail vein, blood was withdrawn at the same time as urine collection. All of the samples were frozen at −30 °C for measurement of urea, albumin, and creatinine levels later on. Seventy-two hours post-BUO, the animals were anesthetized using an intraperitoneal injection of pentobarbitone (45 mg/kg), and the kidneys were collected and stored in either liquid nitrogen, then at −80°C, or in formalin for further assays.

### 2.5. Gene Expression Analysis

A wedge from the middle part of the left kidney, which contains both the cortex and the medulla, was obtained, snap-frozen in liquid nitrogen, and stored at −80 °C for the later measurement of the gene expression of the following by reverse transcription polymerase chain reaction (RT-PCR):Acute kidney injury markers: kidney injury molecule-1 (KIM1) and neutrophil gelatinase-associated lipocalin (NGAL).Cytokines involved in inflammation and fibrosis: tumor necrosis factor-α (TNF-α), interlukin-1 beta (IL-1β), Interleukin-6 (IL-6), and plasminogen activator inhibitor-1 (PAI-1).Pro-apoptotic gene *p53*.Oxidative stress markers: glutathione peroxidase (GPX-1) and glutathione-disulfide reductase (GSR).

The extraction of the total RNA from the frozen samples was performed using the Qiazol Lysis reagent (Qiagen, Hilden, Germany) as per the manufacturer’s protocol. The estimation of the quantity and quality of the extracted RNA was performed using a NanoDrop 2000 Spectrophotometer (Thermo Fisher Scientific Inc, Wilmington, DE, USA).

The preparation of the strand complementary DNA (cDNA) in duplicates from 1.0 µg of extracted RNA was achieved using a QuantiTect^®^ reverse transcription kit (Qiagen, Hilden, Germany) as per the manufacturer’s protocol. The protocol consisted of genomic DNA removal using the supplied gDNA wipeout buffer, ensuring the elimination of interference by genomic DNA. Subsequently, the prepared cDNA was used as a template for relative gene expression analysis using Taqman^®^ hydrolysis probe chemistry. The reaction mixture contained 75 ng cDNA, TaqMan universal master mix (Thermo Fisher Scientific Inc, Wilmington, DE, USA), 0.5 µM of forward and reverse primers, and 0.25 µM of the fluorescent probe (Biosearch Technologies Inc., Petaluma, CA, USA). The probes were FAM-labeled.

The Peptidylprolyl Isomerase A (PPIA) house-keeping gene was used for normalization. Its probe was labeled with Quasar 670, enabling multiplexing with the genes of interest. All of the samples were run in duplicates. At least one primer of all designed PCR primers sets spanned the exon–exon junction to further exclude any interference of the genomic DNA. Table 1 shows the sequences of primers and probes. The calculated cycle threshold (CT) values were used in the estimation of the changes in gene expression of the target genes using the delta–delta CT formula. The cycle threshold (CT) values were used to estimate the delta–delta CT(ΔΔCT) to determine changes in the expression of the target genes.

### 2.6. Histological Studies

The kidney tissue was washed with ice-cold saline and blotted using filter paper, placed in a cassette, and fixed directly in 10% neutral formalin for 24 h. This was followed by dehydration in increasing ethanol concentrations, clearing with xylene, and embedding with paraffin. Then, 3 μm sections were prepared from paraffin blocks and stained with hematoxylin and eosin. The stained sections were evaluated blindly using light microscopy.

The microscopic scoring was performed by the measurement of the percentage of the areas that showed morphologic changes (tubular dilatation, tubular atrophy, interstitial fibrosis, and mononuclear cellular infiltrate) in comparison to the total surface area in each sample. The measurement of the frequency of each histological abnormality was performed using Image J software (1.53t version, NIH, Bethesda, MA, USA).

### 2.7. Statistical Analysis

Statistical analysis was performed using SPSS V16.0. The results are expressed as mean ± SEM. A one-way factorial ANOVA was used for the comparison of variables between groups and between different stages (basal, pre-BUO, and post-BUO) within each group. A *p* value of less than 0.05 was considered statistically significant.

## 3. Results

### 3.1. Renal Functions

As demonstrated in Table 2, the basal serum creatinine, serum urea, creatinine clearance, and albumin–creatinine ratio (ACR) were similar in all the groups (*p* > 0.05 for all variables). In addition, there was no difference in these variables between the pre-BUO and basal values in all the groups (*p* > 0.05 for all variables).

In the sham group, the sham manipulation of the ureters had no effect on the renal functional parameters when compared to the pre-IRI values (*p* > 0.05 for all variables). In the BUO group, there was a significant deterioration in the renal functions following BUO (Table 2). For instance, BUO led to a significant deterioration in serum creatinine and creatinine clearance (0.32 ± 0.02 vs. 0.23 ± 0.01 and 0.63 ± 0.07 vs. 1.06 ± 0.06, respectively, *p* < 0.05 for both). Likewise, BUO resulted in an increase in urinary albumin leakage. So, the ACR increased to 108.0 ± 34.0 from 12.0 ± 1.5 in the pre-BUO time point (*p* < 0.05).

As depicted in Table 2, the administration of nerolidol significantly attenuated BUO-induced alterations in all these parameters (*p* < 0.05 for all variables).

### 3.2. Gene Expression Analysis Results

As illustrated in Figure 2, BUO caused a significant increase in the gene expression of KIM-1 and NGAL, as indicated by the comparison between G-BUO and the G-sham (145.1 ± 32.1 vs. 1.1 ± 0.2 (*p* < 0.001) and 18.5 ± 3.7 vs. 1.0 ± 0.1 (*p* < 0.05), respectively. Conversely, nerolidol treatment significantly mitigated this increase (73.1 ± 13.2 vs. 145.1 ± 32.1 and 9.1 vs. 18.5 ± 3.7), respectively (*p* < 0.05 for both).

A similar trend was observed with respect to pro-inflammatory cytokines (Figure 3). For example, ureteral obstruction caused a significant upregulation of the mRNA expression of TNF-α, PAI-1, IL-6, and IL-1β (2.22 ± 0.47 vs. 1.03 ± 0.11, 5.38 ± 0.87 vs. 1.02 ± 0.10, 10.35 ± 1.42 vs. 1.00 ± 0.21 and 1.55 ± 0.07 vs. 0.99 ± 0., respectively, *p* < 0.05 for all). Nerolidol treatment significantly attenuated these alterations in TNF-α, PAI-1, and IL-6 (1.14 ± 0.18 vs. 2.22 ± 0.47, 3.15 ± 0.44 vs. 5.38 ± 0.87, 6.49 ± 0.54 vs. 10.35 ± 1., respectively, *p* < 0.05 for all). It also mitigated the change in IL-1β, but this did not reach statistical significance (1.18 ± 0.18 vs. 1.55 ± 0.07, *p* = 0.07).

Bilateral ureteral obstruction has also caused a significant increase in the gene expression of some pro-apoptotic markers. For instance, the mRNA expression of p53 was significantly elevated in G-BUO compared to G-sham (1.38 ± 0.05 vs. 0.98 ± 0.03, *p* = 0.001) (Figure 4). Nerolidol significantly attenuated this upregulation (1.17 ± 0.09 vs. 1.38 ± 0.05, *p* < 0.05).

A similar trend was observed with respect to oxidative stress markers (Figure 4). The mRNA expression of GSR was significantly higher in G-BUO compared to G-sham (1.21 ± 0.07 vs. 0.92 ± 0.06, *p* < 0.05) (Figure 5). Nerolidol significantly reduced the expression of GSR in G-BUO/NERO compared to G-BUO (0.97 ± 0.06 vs. 1.21 ± 0.07, *p* < 0.05). Moreover, BUO caused a significant increase in the gene expression of GPx-1, as shown by comparing G-BUO to G-sham (1.31 ± 0.010 vs. 1.01 ± 0.05, *p* < 0.05). Nerolidol treatment mitigated this increase, but this did not reach statistical significance (1.06 ± 0.09 vs. 1.31 ± 0.10, *p* = 0.08) (Figure 4).

### 3.3. Histological Findings

Bilateral ureteral obstruction led to significant histological changes in the architecture of the kidneys. The sham control group showed normal kidney architecture and histology (Figure 5A,B). G-BUO, on the other hand, showed significant tubular necrosis and dilated renal tubules with intratubular eosinophilic secretion in 44.2 ± 2.9% of the fields (Figure 5C,D) (*p* < 0.0001 compared to G-sham). However, G-BUO/NR showed significantly less tubular injury, tubular dilatation, and necrosis compared to G-BUO (27.8 ± 1.5%, *p* < 0.001 (Figure 5E,F).

## 4. Discussion

Obstructive uropathy is a common cause of acute kidney injury characterized by the hindrance of normal urine flow through the urinary tract, resulting from various structural and functional etiologies [22,23,24]. It affects all age groups and can lead to detrimental consequences if left untreated. Our results suggested that the administration of nerolidol can ameliorate the alterations in renal functions, the gene expression levels of key kidney injury markers, pro-inflammatory and pro-apoptotic cytokines, oxidative stress markers, and kidney histology induced by BUO injury Figure 6.

In the current study, we have shown that nerolidol has anti-inflammatory effects on the kidney following BUO. It has mitigated alterations in TNF-α, IL-1β, IL-6, and PAI-1. TNF-α is a versatile pro-inflammatory cytokine that is produced by macrophages and monocytes during acute inflammation [25]. It is a versatile cytokine that is essential in various cellular processes, including cell survival, proliferation, differentiation, and cell death [26]. Its primary role is to mount and regulate the immune response by triggering the production of other important pro-inflammatory cytokines, including IL-1β and IL-6 [27,28]. However, the overexpression of TNF-α due to severe injury could lead to detrimental consequences. Our gene expression analysis revealed that nerolidol significantly attenuated the BUO-induced excessive upregulation in the gene expression of TNF-α in G-BUO/NERO. The exact mechanism for this attenuation is difficult to ascertain from the current study, but could possibly be attributed to the inhibition of TLR4 stimulation by nerolidol, which has been previously demonstrated in other organs [29]. The ability of nerolidol to target the TLR4/NF-κB signaling cascade might interfere with the translocation of phosphorylated nuclear factor-κB (p-NF-κB), an important agent which triggers the synthesis of pro-inflammatory cytokines including TNF-α [15,30]. Regardless of the exact mechanism of the effect of nerolidol on TNF-α changes, the findings of this study are consistent with the findings reported in other models of acute kidney injury, such as lipopolysaccharide- and ischemia–reperfusion-induced acute kidney injury [15,17].

Nerolidol has also attenuated BUO-induced upregulation in IL-6 and IL-1β. The former is a major pro-inflammatory cytokine that plays a key role in the recruitment and interstitial infiltration of neutrophils, thereby exacerbating renal injury. IL-1β is also a key mediator of inflammation that typically spikes during acute renal injury and causes systemic inflammation by activating all types of leukocytes via its receptor, IL-1R [31]. The downregulation in the gene expression of both IL-6 and IL-1β could be due to the inhibition of TNF-α gene expression, as TNF-α is required to trigger the synthesis and release of these cytokines [27,28]. These findings are consistent with previous reports on the effect of nerolidol in other injury models [15,17,32] and indicate that nerolidol exhibits potent anti-inflammatory properties.

Tissue hypoxia is among the pathological consequences of ureteral obstruction [33]. These stress stimuli cause kidney cells to generate and release reactive oxygen species, which in turn exacerbates the renal injury. In the current study, we measured the gene expression level of two important oxidative stress markers, GPx-1 and GSR. The latter, which accounts for 96% of all kidney GPx activity [34], exerts its antioxidant effects by enzymatically reducing hydrogen peroxide to water and oxidizing reduced glutathione (GSH) to glutathione disulfide [35]. In contrast, GSR reduces oxidized glutathione disulfide to the sulfhydryl group forming glutathione (GSH), which plays a pivotal role in the cellular antioxidant defense system [36]. Our results showed that GPx-1 and GSR gene expression was significantly altered by BUO injury, and that nerolidol had attenuated the overexpression of these enzymes. This is probably due to its reactive oxygen species scavenging properties, as nerolidol has been shown to neutralize free radicals in an in vitro model [37]. This reduction in GSR by nerolidol might have an effect on the long-term outcome, as it has been demonstrated that upregulation in GSR expression could lead to collagen type I and IV buildup, which could potentially exacerbate long-term renal damage [38].

In terms of apoptotic markers, our data have revealed that nerolidol successfully attenuated excessive upregulation in the gene expression of p53, a key regulator of cell division and apoptosis. This could be, at least partially, due to nerolidol’s antioxidant properties, as it has been proposed that reactive oxygen species might act, directly or indirectly, as an upstream activator of p53 [39]. Likewise, nerolidol significantly attenuated alteration in the expression of PAI-1, which plays an important role in regulating cellular proliferation and apoptosis.

Owing to nerolidol’s antioxidant, anti-inflammatory, and anti-apoptotic properties, BUO-induced alterations in kidney function were partially mitigated, as shown by the improvement in serum creatinine, serum urea, and creatinine clearance, along with a reduction in the albumin–creatinine ratio. The protective effects of nerolidol extended beyond glomerular functions, including renal tubular functions as well. Gene expression analysis revealed a significant attenuation in the expression levels of two important acute renal injury markers: KIM-1, which is upregulated and released by proximal renal tubules in response to acute kidney injury [40], and NGAL, which is produced in the ascending limb of the loop of Henle and the collecting ducts [41]. Therefore, the mitigation of the upregulation of these two markers by nerolidol indicates that this agent has a protective effect on both proximal and distal tubules. These findings are corroborated by histological results, as the nerolidol-treated group exhibited a significantly improved histological architecture compared to the BUO group, with less tubular damage, focal tubular dilatation, and leukocyte infiltration.

The BUO injury rat model utilized in this study is similar to the clinical scenarios of uropathy observed in some common medical conditions, such as acute bilateral ureteric obstruction due to conditions such as acute urinary retention due to benign prostatic hyperplasia. Thus, the reno-protective effects of nerolidol demonstrated in this study may have significant clinical implications. Further clinical research is necessary to validate and extend these findings to clinical settings.

## 5. Conclusions

The findings of this study demonstrated the potent reno-protective effects of an oral single daily dose of 200 mg/kg of nerolidol in mitigating the adverse renal effects of bilateral ureteral obstruction. The protective effect demonstrated in this study is attributed to its anti-inflammatory, anti-fibrotic, anti-apoptotic, and antioxidant properties. These effects were reflected in the partial recovery of renal functions and histological features. Given the clinical relevance of bilateral ureteral obstruction, the current study results provide a promising basis for future clinical research to explore nerolidol’s potential as a novel treatment in renal injury associated with obstructive uropathy.

## Figures and Tables

**Figure 1 biomedicines-12-02285-f001:**
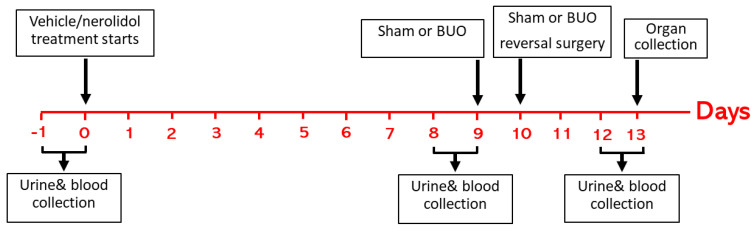
Schematic presentation of the study plan showing interventions in all groups.

**Figure 2 biomedicines-12-02285-f002:**
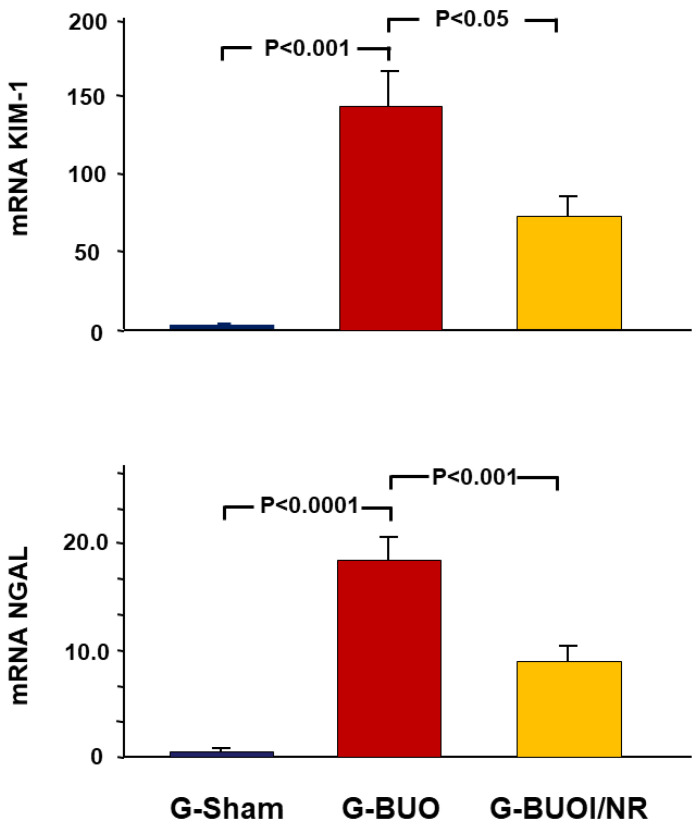
The gene expression of two of markers of acute renal injury (KIM-1 and NGAL) in all the groups. The values represent mean ± SEM.

**Figure 3 biomedicines-12-02285-f003:**
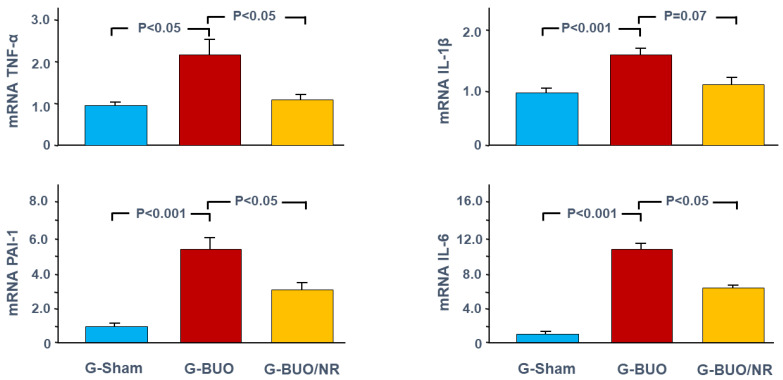
The gene expression of pro-inflammatory markers: tumor necrosis factor-α (TNF-α), plasminogen activator inhibitor-1 (PAI-1), interleukin 1 beta (IL-1β), and interleukin-6 (IL-6) in all groups. The values represent mean ± SEM.

**Figure 4 biomedicines-12-02285-f004:**
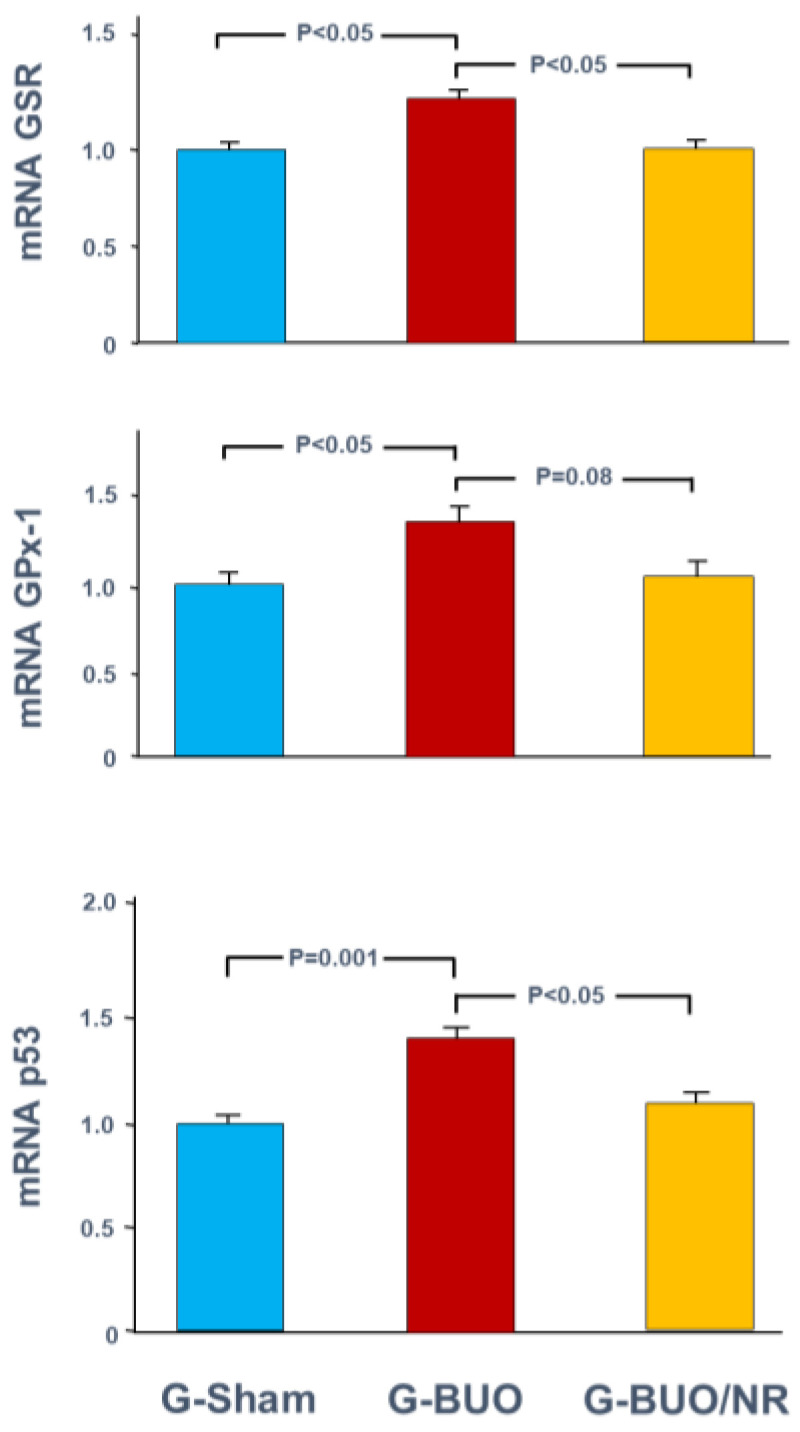
The gene expression of the oxidative stress markers: glutathione peroxidase (GPX-1) and glutathione-disulfide reductase (GSR) pro-apoptotic *p53* gene in all groups. Values represent mean ± SEM.

**Figure 5 biomedicines-12-02285-f005:**
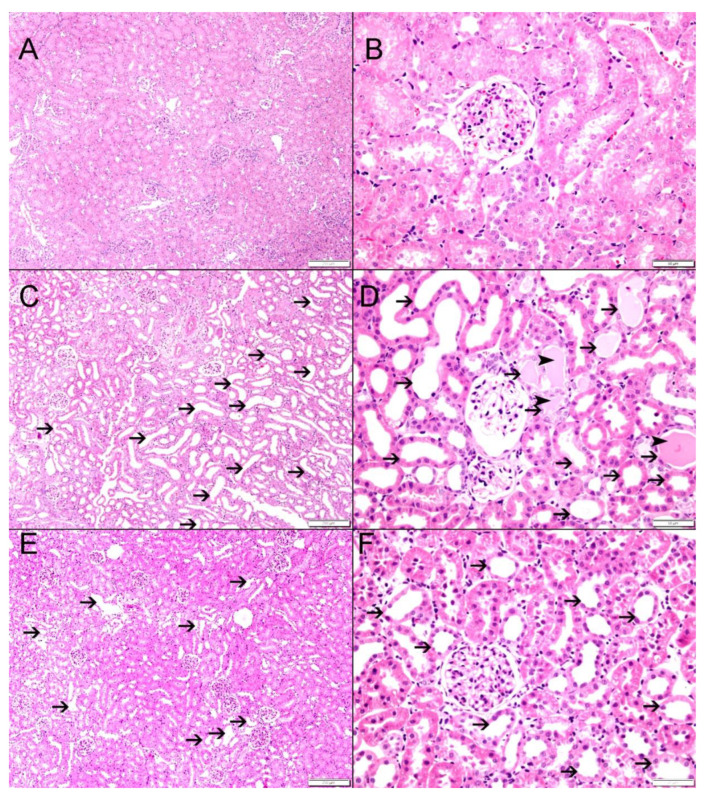
The histological features in all the experimental groups. (**A**,**B**): G-sham control showing normal kidney architecture and histology. (**C**,**D**): G-BUO showing large areas of tubular injury with tubular dilatation (thin arrows) and intratubular secretions (arrowheads) in 44.2 ± 2.9% of the examined tissue areas. (**E**,**F**): G-BUO/NR-treated group showing foci of tubular injury with tubular dilatation (thin arrows) involving 27.8 ± 1.5% of examined tissue areas.

**Figure 6 biomedicines-12-02285-f006:**
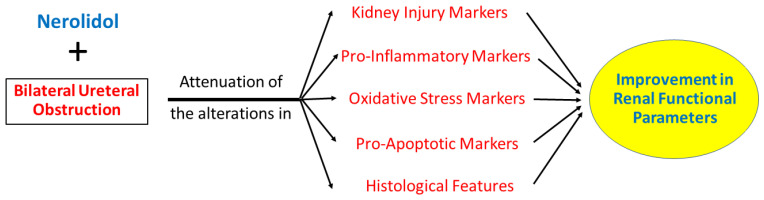
Summary of the effects of nerolidol on bilateral ureteral obstruction.

**Table 1 biomedicines-12-02285-t001:** Forward and reverse primers and fluorogenic probe sequences used for real-time quantitative PCR analysis. KIM-1: kidney injury molecule-1; NGAL: neutrophil gelatinase-associated lipocalin; TNF-α: tumor necrosis factor-alpha; PAI-1: plasminogen activator inhibitor-1; IL-6: interleukin 6; IL-1β: interleukin 1 beta; p53: pro-apoptotic gene *p53*; GSR: glutathione-disulfide reductase; GPx-1: Glutathione peroxidase 1; PPIA: peptidylprolyl isomerase A (housekeeping gene).

KIM-1 (NM_173149.2)	Forward	GCCTGGAATAATCACACTGTAAG
	Reverse	GCAACGGACATGCCAACATAG
	Probe	d FAM-TCCCTTTGAGGAAGCCGCAGA-BHQ-1
NGAL (NM_130741.1)	Forward	CTGTTCCCACCGACCAATGC
	Reverse	CCACTGCACATCCCAGTCA
	Probe	FAM-TGACAACTGAACAGACGGTGAGCG-BHQ-1
TNF-α (NM_012675.3)	Forward	CTCACACTCAGATCATCTTCTC
	Reverse	CCGCTTGGTGGTTTGCTAC
	Probe	FAM-CTCGAGTGACAAGCCCGTAGCC-BHQ-1
PAI-1 (NM_012620.1)	Forward	GGCACAATCCAACAGAGACAA
	Reverse	GGCTTCTCATCCCACTCTCAAG
	Probe	FAM-CCTCTTCATGGGCCAGCTGATGG-BHQ-1
IL-6 (NM_012589.2)	Forward	TCACAGAGGATACCACCCACAACA
	Reverse	CACAAGTCCGGAGAGGAGAC
	Probe	FAM-TCAGAATTGCCATTGCACAACTCT-BHQ-1
IL-1β (NM_031512.2)	Forward	ATGCCTCGTGCTGTCTGACC
	Reverse	GCTCATGGAGAATACCACTTGTTGG
	Probe	FAM-AGCTGAAAGCTCTCCACCTCAATGGA-BHQ-1
p53 (NM_030989.3)	Forward	CGAGATGTTCCGAGAGCTGAATG
	Reverse	GTCTTCGGGTAGCTGGAGTG
	Probe	FAM-CCTTGGAATTAAAGGATGCCCGTGC-BHQ-1
GSR NM_053906.2	Forward	CATCCCTACCGTGGTCTTCAG
	Reverse	ATGGACGGCTTCATCTTCAGT
	Probe	FAM-CCACCCGCCTATCGGGACAGT-BHQ-1
GPx-1 NM_030826.4	Forward	GTGCTGCTCATTGAGAATGTCG
	Reverse	TCATTCTTGCCATTCTCCTGATG
	Probe	FAM-TCCCTCTGAGGCACCACGAC-BHQ-1
PPIA (NM_017101.1)	Forward	GCGTCTGCTTCGAGCTGT
	Reverse	CACCCTGGCACATGAATCC
	Probe	Quasar 670-TGCAGACAAAGTTCCAAAGACAGCA-BHQ-2

**Table 2 biomedicines-12-02285-t002:** Renal functional parameters in all the groups; basal: before the start of treatment; pre-BUO: just prior to the intervention (BUO/sham surgery), and post-BUO: 3 days after reversal of BUO/sham surgery. The results represent mean ± SEM; ^$^ indicates statistical significance when compared to the sham group (G-sham); and * indicates statistical significance when compared to G-BUO.

	G-sham			G-BUO			G-BUO/NR		
	Basal	Pre-BUO	Post-BUO	Basal	Pre-BUO	Post-BUO	Basal	Pre-BUO	Post-BUO
Serum Creatinine (mg/dL)	0.20 ± 0.01	0.22 ± 0.01	0.21 ± 0.02	0.19 ± 0.01	0.23 ± 0.01	0.32 ± 0.02 ^$^	0.19 ± 0.01	0.21 ± 0.01	0.26 ± 0.02 *
Serum Urea (mg/dL)	46.9 ± 1.3	44.7 ± 1.7	43.5 ± 2.8	48.0 ± 1.2	44.6 ± 1.2	56.5 ± 1.7 ^$^	45.9 ± 2.9	45.5 ± 1.9	50.2 ± 1.1 *
Creatinine Clearance (mL/min/100 gm b.w.)	1.01 ± 0.09	1.04 ± 0.08	1.08 ± 0.07	1.03 ± 0.06	1.06 ± 0.06	0.63 ± 0.07 ^$^	0.99 ± 0.07	1.09 ± 0.09	0.95 ± 0.11 *
Albumin Creatinine Ratio	13.8 ± 1.9	11.4 ± 2.7	14.1 ± 1.5	15.9 ± 2.6	12.0 ± 1.5	180.0 ± 34.0 ^$^	15.0 ± 4.1	11.8 ± 2.7	73.5 ± 18.9 *

## Data Availability

Data is available on request due to technical issues.

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
