# Peer review of "The Effect of Nerolidol on Renal Dysfunction following Bilateral Ureteral Obstruction"

_biomedicines, 2024, doi:10.3390/biomedicines12102285_

Round 1

Reviewer 1 Report

Comments and Suggestions for Authors The objective of this study was to examine the impact of nerolidol in a rat model of reversible bilateral ureteral obstruction (BUO). The rats were randomly allocated to three groups: G-Sham, G-BUO, and G-BUO/NR. Renal functions were evaluated prior to the initiation of treatment, immediately before the intervention, and three days following the reversal of BUO. Neither nerolidol nor the vehicle influenced the basic renal functions. Nerolidol significantly mitigated the BUO-induced changes in renal functional measures, including serum creatinine, urea, creatinine clearance, and urine albumin-creatinine ratio. Nerolidol also mitigated alterations in many indicators linked to kidney damage, inflammation, apoptosis, and oxidative stress. My first concern is that the introduction is not well referenced. The author miss some key citations in the introduction section (https://doi.org/10.1080/10717544.2023.2241661 and other) The overall methodology adopted in the manuscript (figure 1) can be revised for clarity. Figures 1-6 can be merged as panels, so that it is easy for comparison. In “discussion” the authors could add a figure or table summarizing the effects of nerolidol I suggest the authors add a conclusion section separately and it must be more informative and unveil the most beneficial outcomes.

Author Response

Index of Changes- Manuscript #: ID: biomedicines-3231478

The effect of nerolidol on the renal dysfunction following bilateral ureteral obstruction

The authors would like to thank the Editor and the Reviewers for the positive critical comments which have strengthened the manuscript.

In this document, we have responded to the Reviewers’ comments. The changes refer to the highlighted revised version. In the highlighted version of the manuscript, the underlined text indicates that it has been added whereas strikethrough sign indicates deletion of the text. All modifications are indicated in red. A final neat version was also included.

In addition to addressing the Reviewer’s comments, we made further modifications in the text to render it easier to understand and follow by the reader (Page: 2, Line: 39 and 45), (Page: 3, Line: 51, 54-56, 63), (Page: 13, Line: 252, 256), (Page: 14, Line: 273, 275, 276, 279, 281, 283), (Page: 15, Line: 284, 292-293, 295).

Reviewer#1:

The objective of this study was to examine the impact of nerolidol in a rat model of reversible bilateral ureteral obstruction (BUO). The rats were randomly allocated to three groups: G-Sham, G-BUO, and G-BUO/NR. Renal functions were evaluated prior to the initiation of treatment, immediately before the intervention, and three days following the reversal of BUO. Neither nerolidol nor the vehicle influenced the basic renal functions. Nerolidol significantly mitigated the BUO-induced changes in renal functional measures, including serum creatinine, urea, creatinine clearance, and urine albumin-creatinine ratio. Nerolidol also mitigated alterations in many indicators linked to kidney damage, inflammation, apoptosis, and oxidative stress. 

Comment#1: My first concern is that the introduction is not well referenced. The author misses some key citations in the introduction section (https://doi.org/10.1080/10717544.2023.2241661 and other).

Response: Several changes has been made in the Introduction to address these points with addition of more references including the one suggested by the Reviewer (Page: 3, Line: 53-58, 61, 72-73, 75-77), (Page: 4, Line: 78-89).

Comment#2: The overall methodology adopted in the manuscript (figure 1) can be revised for clarity.

Response: Figure-1 has now been modified and many words have been deleted to simplify the methodology protocol (Page: 21).

Comment#3: Figures 1-6 can be merged as panels, so that it is easy for comparison. 

Response: In the revised version, we tried to merged cytokines with similar functions in one figure; so, we ended up having one figure for the kidney injury markers, one for the pro-inflammatory and pro-fibrotic cytokines and one for both the oxidative stress markers and apoptotic markers. Including all these cytokines and markers in one figure would result in a very small font that will be very difficult for the reader to follow. Further, in the final print version of the journal these figures will supposedly be in the Results section which means, they will be in one page or two subsequent pages at the maximum. So, we now have Figure-2, Figure-3 and Figure-4 instead of the previous arrangement (Figure-2, Figure-3, Figure-4, Figure-5 and Figure-6 (Pages: 22, 23, 24, 25, 26, 27 and 28).

Comment#4: In “discussion” the authors could add a figure or table summarizing the effects of nerolidol

Response: This has now been added (Page: 13, Line: 248) and Page: 30 (Figure-6).

Comment#5: I suggest the authors add a conclusion section separately and it must be more informative and unveil the most beneficial outcomes.

Response: A separate Conclusion section which is more informative has now been added (Page: 16-17, Line: 324-336).

Reviewer 2 Report

Comments and Suggestions for Authors

The paper studies the effect of nerolidol on renal dysfunction, and the experimental design is generally reasonable. Therefore, I accept the paper for publication after addressing the following major concerns.

(1)The introduction discusses bilateral ureteral obstruction (BUO) and its effects on acute kidney injury. But the potential of natural products to treat renal dysfunction is not discussed in detail. And the comparisons with other similar studies could be included.

(2)The results section provides a rather simplistic description of certain gene expression changes. Further expansion of the discussion on key genes such as KIM-1, NGAL, and TNF-α is needed.

(3)The consistency or discrepancies of the experimental results with existing literature need to be discussed and explained. 

(4)Some minor mistakes including some grammar should be checked and corrected.

Comments on the Quality of English Language

Some minor mistakes including some grammar should be checked and corrected.

Author Response

Index of Changes- Manuscript #: ID: biomedicines-3231478

The effect of nerolidol on the renal dysfunction following bilateral ureteral obstruction

The authors would like to thank the Editor and the Reviewers for the positive critical comments which have strengthened the manuscript.

In this document, we have responded to the Reviewers’ comments. The changes refer to the highlighted revised version. In the highlighted version of the manuscript, the underlined text indicates that it has been added whereas strikethrough sign indicates deletion of the text. All modifications are indicated in red. A final neat version was also included.

In addition to addressing the Reviewer’s comments, we made further modifications in the text to render it easier to understand and follow by the reader (Page: 2, Line: 39 and 45), (Page: 3, Line: 51, 54-56, 63), (Page: 13, Line: 252, 256), (Page: 14, Line: 273, 275, 276, 279, 281, 283), (Page: 15, Line: 284, 292-293, 295).

Reviewer #2:

The paper studies the effect of nerolidol on renal dysfunction, and the experimental design is generally reasonable. Therefore, I accept the paper for publication after addressing the following major concerns.

Comment#1: The introduction discusses bilateral ureteral obstruction (BUO) and its effects on acute kidney injury. But the potential of natural products to treat renal dysfunction is not discussed in detail. And the comparisons with other similar studies could be included.

Response: An extra text with more references has been now been added to address this point (Page: 3, Line: 53-58, 61, 72-73, 75-77), (Page: 4, Line: 78-89).

Comment#2: The results section provides a rather simplistic description of certain gene expression changes. Further expansion of the discussion on key genes such as KIM-1, NGAL, and TNF-α is needed.

Response: An extra text has been added and some changes were made to address this point (Page: 13, Line: 252, 256, 259-261), (Page: 14, Line: 262-268, 273, 275, 276-279, 281, 283), (Page: 15, Line: 284, 292-295), (Page: 16, Line: 312-313).

Comment#3: The consistency or discrepancies of the experimental results with existing literature need to be discussed and explained.

Response: In addition to the existing text, an extra text has been added to address this point (Page: 14, Line: 265-268, 276-278).

Comment#4: Some minor mistakes including some grammar should be checked and corrected.

Response: The whole manuscript has now been re-checked for any grammatical errors (Page: 2, Line: 39 and 45), (Page: 3, Line: 51, 54-56, 63), (Page: 13, Line: 252, 256), (Page: 14, Line: 273, 275, 276, 279, 281, 283), (Page: 15, Line: 284, 292-293, 295).

Round 2

Reviewer 2 Report

Comments and Suggestions for Authors

The article has been revised as required, and I agree that the article is published in the journal Biomedicines